# A systematic review of fuzzing based on machine learning techniques

**Yan Wang[1], Peng Jia[1], Luping Liu[2], Cheng Huang[1], Zhonglin Liu[1]***

**1** College of Cybersecurity Sichuan University, Chengdu, China, **2** College of Electronics and Information Engineering, Sichuan University, Chengdu, China

* jungleforsa@gmail.com

## Abstract

Security vulnerabilities play a vital role in network security system. Fuzzing technology is widely used as a vulnerability discovery technology to reduce damage in advance. However, traditional fuzz testing faces many challenges, such as how to mutate input seed files, how to increase code coverage, and how to bypass the format verification effectively. Therefore machine learning techniques have been introduced as a new method into fuzz testing to alleviate these challenges. This paper reviews the research progress of using machine learning techniques for fuzz testing in recent years, analyzes how machine learning improves the fuzzing process and results, and sheds light on future work in fuzzing. Firstly, this paper discusses the reasons why machine learning techniques can be used for fuzzing scenarios and identifies five different stages in which machine learning has been used. Then this paper systematically studies machine learning-based fuzzing models from five dimensions of selection of machine learning algorithms, pre-processing methods, datasets, evaluation metrics, and hyperparameters setting. Secondly, this paper assesses the performance of the machine learning techniques in existing research for fuzz testing. The results of the evaluation prove that machine learning techniques have an acceptable capability of prediction for fuzzing. Finally, the capability of discovering vulnerabilities both traditional fuzzers and machine learning-based fuzzers is analyzed. The results depict that the introduction of machine learning techniques can improve the performance of fuzzing. We hope to provide researchers with a systematic and more in-depth understanding of fuzzing based on machine learning techniques and provide some references for this field through analysis and summarization of multiple dimensions.

**Data Availability Statement:** All the files are available from the certain Github database (https://github.com/wtwofire/A-systematic-review-of-fuzzing-based-on-machine-learning-techniques).

## Introduction

Vulnerabilities often refer to flaws or weaknesses in hardware, software, protocol implementations, or system security policies that allow an attacker to access or compromise the system without authorization, and have become the root cause of the threats toward network security. The number of vulnerabilities announced by CVE (Common Vulnerabilities and Exposures) began to explode in 2017, from the original highest 7946 vulnerabilities in 2014 to the publication of 16556 vulnerabilities in 2018 [1]. In recent cybersecurity incidents, WannaCry

**Funding:** The author(s) received no specific funding for this work.

**Competing interests:** The authors have declared that no competing interests exist.

ransomware attacking outbroke on May 2017, and more than 150 countries and 300,000 users were attacked, causing more than $8 billion in damage [2]. The virus spread widely by utilizing the "Eternal Blue" vulnerability of the NSA (Nation Security Agency) leak.

Considering the severe damages caused by vulnerabilities, vulnerability discovery technology has attracted widespread attention. Fuzzing in vulnerability discovery techniques is an efficient method to discover the software weaknesses, which was first proposed by Miller et al. [3]. It is an automatic testing technique that covers numerous boundary cases using invalid data (e.g., files, network packets, program codes) as program input to discover more vulnerabilities [4]. Since then, a multitude of different techniques were proposed to improve the efficiency of fuzzing. These techniques include static analysis [5, 6] and dynamic analysis [7–9]. However, fuzz testing still faces many challenges, such as how to mutate seed inputs, how to increase code coverage, and how to bypass verification effectively [10].

Machine learning has been used in the field of cybersecurity, and it has also been adopted by many studies for vulnerability detection [11–13], including the applications in fuzzing [14–17]. The combination of fuzz testing and machine learning techniques provides a new idea to alleviate the bottleneck problem of traditional fuzzing techniques, which also makes the fuzz testing process intelligent. Machine learning techniques used in fuzz testing will become one of the key points in the development of vulnerability detection techniques with the explosive growth of machine learning research.

However, there is no systematic review of machine learning based fuzzing in the past few years. We firmly believe that it is necessary to write a comprehensive review to summarize the latest methods and new research results in this area. The paper aims to discuss, analyze, and summarize the following problems:

- RQ1: Why machine learning techniques can be used for fuzzing?

- RQ2: Which steps in the fuzzing have used machine learning techniques?

- RQ3: Which machine learning algorithms have been used for fuzzing?

- RQ4: Which techniques are used for data pre-processing of fuzzing based on machine learning?

- RQ5: Which datasets are used for training and evaluating?

- RQ6: Which performance measures are used for evaluating the results?

- RQ7: How to set the hyperparameters of the machine learning arithmetic?

- RQ8: What is the performance of the machine learning arithmetics?

- RQ9: What is the capacity of the machine learning techniques based fuzzers to discover vulnerabilities?

The rest of the paper is organized as follows: Section 2 provides the background to this review. Section 3 presents the research criteria followed in this study for selection of primary studies. Section 4 analysis the extracted data by using a narrative format according to pre-determined themes emerged from the research questions. Section 5 presents the answers to the research questions identified in this work. Section 6 provides the limitation of this work and Section 7 provides conclusions and future directions obtained from this systematic review.

## Background

**Working process of fuzzing.** The processes of traditional fuzzing are depicted in Fig 1. Some steps in the workflow, such as testcase filters, may not be needed in some fuzzers, but the

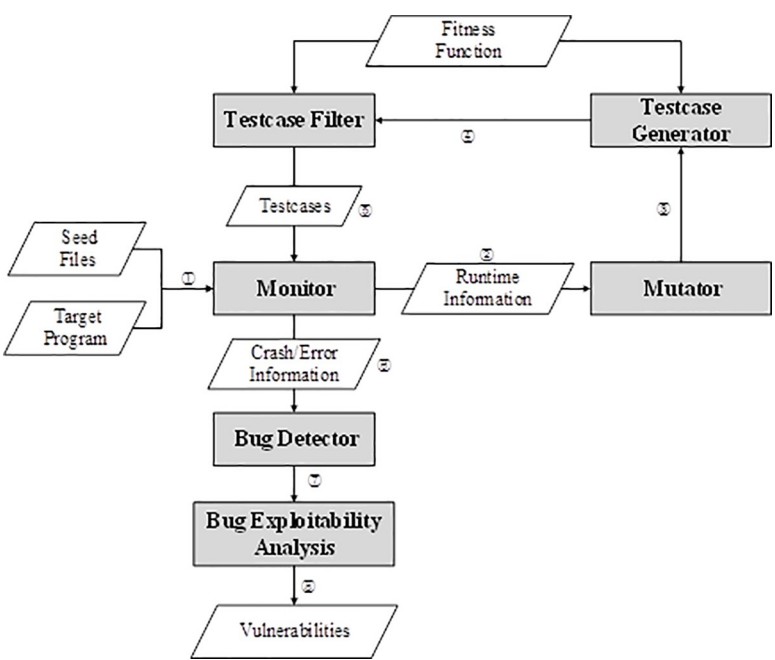

**Fig 1. Working process of fuzzing.**

general steps are similar. The working process of fuzzing is composed of four main stages: testcase generation, program execution, runtime state monitoring, and analysis of crashes.

The testcase generation stage primarily provides input for the fuzzing process. It includes seed file generation, mutation, testcase generation, and testcase filtering. The seed files are the original samples that conform to the format of the program's inputs. The mutation of seed files can generate a host of testcases by selecting different mutation strategies in different locations [18]. An essential function of the testcase filtering is to select testcases that can trigger new paths or vulnerabilities because not all testcases are valid. The selection process is guided by the defined fitness functions, which refers to the evaluation methods used to distinguish the satisfactory (e.g., code coverage has increased) and unsatisfactory (e.g., code coverage has not increased) standards of testcase in the state-of-the-art fuzzer based on genetic algorithms. The testcase generation stage has two kinds of generators: mutation-based generators and generation-based generators. Mutation-based testcase generation strategies generate new testcases by modifying known seed files. Generation-based testcase generation strategies generate new testcases based on the format information of the input sample without mutation.

The program execution stage is mainly to feed the generated testcases into the target program for execution. The program under test (PUT) can be executable binary codes with or without source codes.

The runtime state monitoring stage monitors the state of the program at runtime. Then, the information monitored at this stage will be fed back into the testcase generation stage to guide the next generation of testcase [19, 20]. The techniques used in the monitor include binary code instrumentation [21] and taint analysis [22]. When a target program crashes or reports some errors, the related information will be collected for later replay and analysis.

In the analysis stage, the information collected from the crashes will be analyzed to determine whether a crash is a bug or not. Then, fuzzer classify bugs by exploitability analysis to determine whether it is a vulnerability or not [23–25]. Finally, the analysts make the final confirmation by debugging.

**The limitations of fuzzing.** The traditional fuzzer faces three key challenges: (1) how to mutate seed inputs, (2) how to improve code coverage, (3) how to bypass the validation [10]. In the past, various auxiliary analysis techniques were produced to alleviate these challenges. The introduction of these techniques (static analysis and dynamic analysis) has promoted the development of fuzzing techniques and made the traditional fuzzing process more intelligent. However, these techniques still have certain limitations and disadvantages.

Static analysis extracts relevant information from a binary code or source code without running the program [26], thus is lacks context information during the running and has a high dependence on prior knowledge, resulting in low accuracy and a high false-positive rate.

Dynamic analysis needs to execute the target program in real systems or emulators [27]. It monitors program status and obtains relevant runtime information at execution time. Currently, mainstream dynamic analysis techniques include dynamic symbolic execution [28–30] and dynamic taint analysis [22, 31, 32].

Dynamic symbolic execution collects a set of path constraints by using symbolic values as input during program execution [33]. Then the path constraints will be solved by satisfying model theory (SMT) solver to determine whether the path is reachable or not. If the path is reachable, a corresponding test input will be generated. However, there are many limitations on symbolic execution, such as path explosion [34], environment interactions [35, 36], memory modeling [37, 38], and parallel computing [39, 40].

Dynamic taint analysis uses the tagged data as input, then records how the program uses the input data and which program elements were tainted. However, taint analysis suffers from the problems of under-tainting and over-tainting [41, 42].

Considering the challenges of traditional fuzzing techniques and the shortcomings of the various assistive techniques, fuzz testing requires a combination of new techniques and methods to mitigate these challenges.

# Methods

## Inclusion and exclusion criteria

To keep our survey of the state-of-the-art focused and manageable, it is crucial to define what is and what is not within scope. Overall, the main restriction is that we focus on works related to machine learning based fuzzing. This restriction, in turn, introduces the following five constraints:

1. We exclude fuzzing approaches that don't use machine learning.

2. We exclude approaches that combine machine learning to discover vulnerabilities without use fuzzing.

3. We limit the survey to papers published on peer-reviewed venues and technical reports from academic institutions. Thus, we do not analyze tools, but rather the research works describing their approach.

4. The searches were limited by publication date from 2010 to 2020, and the included studies were limited to the English language.

5. not accessible via the Web.

## Source material and search strategy

To identify candidate papers, we first systematically examined all papers published in 15 top venues for computer security, software engineering, and artificial intelligence from 2010 to 2020: IEEE S&P, ACM CCS, USENIX Security, NDSS, ACSAC, RAID, ESORICS, ASIACCS,

DIMVA, ICSE, FSE, ISSTA, ASE, MSR, and AAAI. To identify candidate papers in other venues, we extensively queried specialized search engines such as Google Scholar. We also carefully examined the references of the candidate papers for any further papers we may have missed.

Words related to fuzzing include: fuzz, fuzzing, fuzzer, input generation, test case, seed generation, crashes exploitability, mutation, and words related to machine learning include: machine learning, neural network, deep learning, reinforcement learning, generative adversarial network, embedding, bayesian network, decision tree, support vector machine, genetic algorithms, random forest. We formed the sophisticated search terms by incorporating alternative terms and synonyms using Boolean expression 'OR' and combining main search terms using 'AND'. For example, the following general search terms were used for identification of primary studies:

Fuzzing AND machine learning, fuzzing AND neural network, fuzzer AND deep learning, fuzzing AND reinforcement learning, fuzzing AND embedding, fuzzing AND bayesian network, fuzzing AND decision tree, fuzzing AND support vector machine, fuzzing AND genetic algorithms, fuzzing AND random forest, and other similar combinations.

## Study selection

After performing the search, all studies were entered into a Reference Manager System (EndNote) and duplicates were removed. Then, the titles and abstracts of the remaining studies were screened using the inclusion and exclusion criteria. Where a decision about inclusion could not be made, the full paper was read to make a definitive judgement.

## Data extraction

In this step, data was extracted from the included studies using a data extraction form. The form was developed specifically for this review and was piloted on a sample of five papers. The form included seven items as shown in Table 1.

## Synthesis of results

After data extraction, data analysis of the studies was performed. The extracted data were analyzed using a narrative format according to pre-determined themes emerged from the research questions. The following themes:

1. Introduction of machine learning technology.

**Table 1. Data extraction form items.**

| Data item | Descript |
|---|---|
| Reference | Title, author, type(eg. Conference/Workshop/journal), data |
| Step | Machine learning is used in the steps of fuzzing |
| Arithmetic | Machine learning arithmetic applied in the study |
| Hyperparameters | The value of the hyperparameter required for machine learning |
| Pre-processing | Pre-processing methods for machine learning techniques in fuzzing |
| Dataset | Source of dataset used to train machine learning models |
| Evaluation | Description of how the machine learning based fuzzer was evaluated |
| Outcome | Results of the evaluation |
| Type | The type of the fuzzer |
| Comments | Remark about the study quality |

2. Applying machine learning techniques to different fuzzing steps.

3. Analysis of machine learning based fuzzing model.

4. Performance evaluation of fuzzing model based on machine learning.

## Results

A total 172 of the most records were retrieved from the 15 top venues, google scholar and the references of the candidate papers. We read each candidate paper to determine if they proposed a fuzzing approach based on machine learning that satisfied the above scope constraints. In the end, a total of 44 papers were selected for this survey, for the full results of the review process, see Fig 2, and the details of the selected papers are listed in Table 2. A standardised approach was used for this systematic review (S1 File). Table 2 provides a unique identifier for each selected primary study along with the reference. These unique identifiers will be used in the tables of all subsequent sections to refer to their corresponding selected primary study.

### Introduction of machine learning technology

Machine learning acquires new knowledge or skills by learning from existing example data or experiences and optimizes the performance of the computer system itself automatically [83]. Machine learning tasks can be categorized into traditional machine learning, deep learning, and reinforcement learning. Traditional machine learning is divided into supervised learning, unsupervised learning, and semi-supervised learning according to whether the input data is labeled or not and the amount of labeled data.

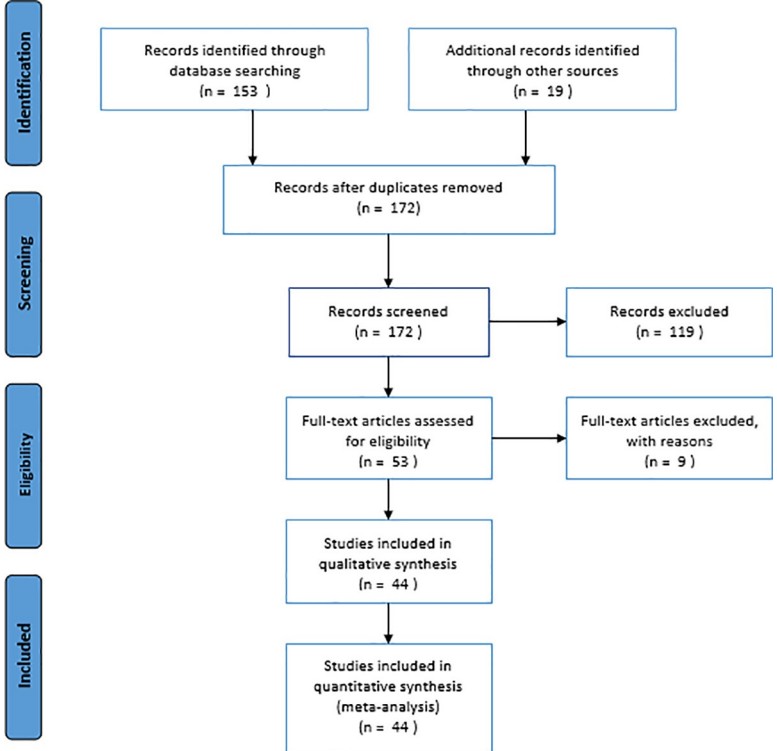

**Fig 2. Flowchart of the systematic review process.**

**Table 2. Selected primary studies in the field of machine learning based fuzzing.**

| Study no. | Paper | Fuzzing Step | Fuzzer Type | Arithmetic | Ref no. | Study no. | Paper | Fuzzing step | Fuzzer Type | Arithmetic | Ref no. |
|---|---|---|---|---|---|---|---|---|---|---|---|
| SP1 | Becker (2010) | MOS | network | *SARSA* | [43] | SP23 | Zhang (2018) | EA | software | *RF, PA* | [62] |
| SP2 | Godefroid (2017) | TG | software | *LSTM, Seq2seq* | [14] | SP24 | Jitsunari (2019) | TG | software | *LSTM* | [63] |
| SP3 | Nichols (2017) | SFG | software | *LSTM, GAN* | [44] | SP25 | Li (2019a) | TG | software | *Struc2vec* | [64] |
| SP4 | Rajpal (2017) | TG | software | *LSTM, Seq2seq* | [15] | SP26 | Li (2019b) | TG | network | *WGAN* | [65] |
| SP5 | Gong (2017) | TF | software | *BLSTM, MLP* | [45] | SP27 | Sperl (2019) | TG | cyber-physical system | *KNN* | [66] |
| SP6 | Wang (2017) | SFG | software | *PCSG* | [17] | SP28 | She (2019) | TG | software | *CNN* | [16] |
| SP7 | Yan (2017) | EA | software | *NB, DT* | [46] | SP29 | Liu (2019b) | TG | software | *Seq2seq* | [67] |
| SP8 | Tripathi (2017) | EA | software | *SVM* | [47] | SP30 | Liu (2019c) | MOS | software | *Deep Q-Learning* | [68] |
| SP9 | Raj (2017) | MOS | cyber-physical system | CNN | [48] | SP31 | Wang (2019) | SFG | software | *LSTM* | [69] |
| SP10 | Fan (2018) | TG | network | *LSTM, Seq2seq* | [49] | SP32 | Zhao (2019a) | TG | network | *LSTM, Seq2seq* | [70] |
| SP11 | Hu (2018) | TG | network | *RNN, CNN, GAN* | [50] | SP33 | Zhang (2019) | TG | software | *BN* | [71] |
| SP12 | Lv (2018) | SFG | software | *MLP, WGAN* | [51] | SP34 | He (2019) | TG | software | *GRU, GCN* | [72] |
| SP13 | Karamcheti (2018a) | MOS | software | *Thompson Sampling* | [52] | SP35 | Kuznetsov (2019) | EA | software | *Deep Q-Learning* | [73] |
| SP14 | Karamcheti (2018b) | TF | software | *LR* | [53] | SP36 | Gao (2019) | TG | network | *LSTM, seq2seq-attention* | [74] |
| SP15 | Cummins (2018) | TG | software | *LSTM* | [54] | SP37 | Cheng (2019) | SFG | software | *RNN, Seq2seq* | [75] |
| SP16 | Sun (2018) | TF | software | *PCFG* | [55] | SP38 | Chen (2019) | TF | cyber-physical system | *SVM, LSTM* | [76] |
| SP17 | Böttinger (2018) | MOS | software | *Deep Q-Learning* | [56] | SP39 | Zhao (2019b) | TG | software | *BLSTM* | [77] |
| SP18 | Sablotny (2018) | TG | software | *RNN, LSTM, GRU* | [57] | SP40 | Joffe (2019) | TG | software | *RNN* | [78] |
| SP19 | Drozd (2018) | MOS | software | *LSTM, Deep Double Q-Learning* | [58] | SP41 | Zong (2020) | TF | software | *CNN* | [79] |
| SP20 | Paduraru (2018) | TG | software | *Seq2seq* | [59] | SP42 | Lee (2020) | TG | software | *LSTM* | [80] |
| SP21 | Nasrabadi (2018) | TG | software | *LSTM, BLSTM* | [60] | SP43 | Lai (2020) | TG | network | *RNN* | [81] |
| SP22 | Fang (2018) | MOS | network | *Q-Learning* | [61] | SP44 | Chen (2020) | SFG | software | *RLS* | [82] |

**EA:** Exploitability analysis, **MOS:** Mutation operator selection, **SFG:** Seed file generation, **TF:** Testcase filtering, **TG:** Testcase generation.

Deep learning [84, 85] is an artificial neural network composed of multiple nonlinear processing units for data representation learning, which is a deeper extension of the machine learning algorithm. The original "feature engineering" is replaced based on representational learning, and the machine can automatically extract useful features from the input data. Reinforcement learning [86] is a branch of machine learning methods that describes and solves the problems of agents maximizing feedback or achieving specific goals by learning strategies in the interaction with the environment. Unlike deep learning, which automatically learns characteristics from a multitude of input samples, reinforcement learning is essentially an automatic decision-making process.

Current machine learning techniques have been widely used in statistical learning [87] pattern recognition [88], data mining [89], computer vision [90], and natural language processing [91] In the field of cyberspace security, researchers have also used machine learning for scenarios such as malicious code detection [92, 93], intrusion detection [94, 95], spam and phishing classification [96, 97], and log analysis [98, 99].

The application of machine learning technology on fuzz testing has also attracted the attention of security researchers, and the reasons can be summarized as follows:

- Many stages of the fuzzing process can be reviewed as classification problems, which are ideal for using machine learning algorithms to address, e.g., whether seed files and testcases are valid, the exploitability of crash, and which mutation operators to choose. Machine learning techniques have relatively excellent abilities in dealing with classification problems.

- The time overhead of testcase generation and testcase execution is an essential factor affecting the performance of fuzzing. There are many steps in traditional fuzzing that require manual definition, and this process is time-consuming. However, the prediction process of the machine learning model just consumes less time. The time spent in fuzz testing can be reduced by replacing some of the methods or steps with machine learning techniques,.

- Bypassing the format check of the program is an indispensable means for the fuzzing process to explore more in-depth code. The traditional manual definition of input syntax format is often time-consuming and relatively simple. Machine learning automatically learns grammatical rules from a large pool of samples that conforms to the syntax specifications.

- The definition of vulnerability features often requires expert knowledge, and it is arduous to adequately define general vulnerability rules. Machine learning can be used to extract the hidden vulnerability features from existing numerous vulnerability-related data, which can better guide the fuzzing process.

Although machine learning techniques have many advantages, the use of machine learning techniques in fuzzing still requires certain prior conditions: 1) training requires massive samples, 2) supervised learning requires the labeled data, and 3) the inputs require to be converted to vectors. For the first and second conditions, the fuzzing process is sufficient because the fuzzing can produce a large number of test samples and crash samples, which can be labeled during sample generation (e.g., whether code coverage increases during execution). Since many of the input files of fuzzing can be treated directly as textual data, natural language processing provide an effective means of converting various data into vectors. Therefore, the third prior condition is also satisfied. The satisfaction of these prior conditions and the advantages of machine learning have led to rapid growth in the research of machine learning based fuzzing.

Applying machine learning techniques to different fuzzing steps

We classify these papers into five categories according to the problems they solved:

- Seed file generation

- Testcase generation

- Testcase filtering

- Mutation operator selection

- Exploitability analysis

Table 3 lists the corresponding research articles in each category. Testcase generation is the most frequent step being combined with machine learning techniques, which has 22 research literature. Part of the reason for this is that it is easiest to collect large quantities of samples and label them from the testcase generation step. Second, the quality of the input sample has a more significant impact on the performance of the fuzzing.

## Seed file generation

The seed file is mutated into fuzzing input samples by various mutation operations. The quality of the seed file is an essential factor influencing the testing effectiveness. Providing good seed files could save lots of CPU times consumed by constructing one, and mutation based on well format seed input is more likely to generate testcases that could reach deeper and hard to reach paths. However, the current seed selection strategies (e.g., using standard benchmarks, crawling from the Internet, and using existing POC samples) have shortcomings. For example, they require more time to acquire the seed set, and the execution effect of selected seeds is almost the same as that of randomly selected seeds [100]. To overcome these shortcomings, Skyfire (SP6) used PCFG (Probabilistic context-sensitive grammar, which contains semantic rules and grammatical features) to extract semantic information automatically to generate seed files. Fast fuzzing (SP3) and SmartSeed (SP12) chose to generate valuable seed files by the antagonistic means of Generative Antagonistic Network (GAN) to learn features. The generated seed file will eventually be used as the input for fuzzing, but different seed files will correspond to different execution paths of the program. Cheng et al. (SP37) learned the correlation between PDF files and the execution paths to generate seed files that can explore new paths by using seq2seq model. Especially, Wang et al. (SP31) proposed a vulnerability-oriented model, NeuFuzz, which uses the LSTM model to learn about known vulnerabilities in the samples to discover execution paths that may contain the vulnerabilities. Finally, NeuFuzz prioritizes the seeds of the executable path that is found, and those seeds that contain more execute paths that may contain vulnerabilities will be executed first. In the scenario of seed scheduling strategy in hybrid fuzzing, MEUZZ (SP44) uses Recursive Least Square (RLS) algorithm and a set of static and dynamic features, computed from seeds and individual programs, to predict seed utility and perform seed schedule. This method determines which new seeds are expected to produce better fuzzing yields.

## Testcase generation

Testcases can be generated by performing mutations on seed files or be constructed based on known input file formats. As the input of a fuzzer, the content of a testcase will directly affect whether a bug is triggered or not. Therefore, constructing a testcase with high code coverage or vulnerability orientation can effectively improve the efficiency of vulnerability detection in fuzzer.

**Table 3. Distribution of research literature based on machine learning for different steps of fuzzing.**

| Step | Studies |
| --- | --- |
| Seed file generation | SP3, SP6, SP12, SP31, SP37, SP44 |
| Testcase generation | SP2, SP4, SP10, SP11, SP15, SP18, SP20, SP21, SP24, SP25, SP26, SP27, SP28, SP29, SP32, SP33, SP34, SP36, SP39, SP40, SP42, SP43 |
| Testcase filtering | SP5, SP14, SP16, SP38, SP41 |
| Mutation operator selection | SP1, SP9, SP13, SP17, SP19, SP22, SP30 |
| Exploitability analysis | SP7, SP8, SP23, SP35 |

## Mutation-based testcase generation

The most critical problem to answer in mutation-based testcase generation is to determine where the input samples mutate and how to mutate. Mutations at only a few critical locations affect the control flow of execution. Therefore, it is especially important to locate these critical positions in testcases. How the input samples mutate is described in section mutation operator selection. For the selection of mutation locations, the current evolutionary algorithms are often stuck in fruitless sequences of random mutations. Augmented-AFL (SP4) learns mutation models from previous fuzzing. The learned function is used to predict the heat map of the complete input file in the fuzzing, corresponding to the probability of mutations in each location of the file that results in new code coverage. This heat map can be used to determine the priority of the mutation location. NEUZZ (SP28) proposes a gradient-guided search strategy superior to evolutionary algorithms to address the problem of high-dimensional structure optimization, and they use a smooth approximate gradient (i.e., NN models) to solve the problems of program behaviors contain many discontinuities, plateaus, and ridges. The result of the calculation is the optimal solution for higher coverage. By using this result, it is effective to identify the target mutation location, and then mutate the seed file to generate test cases that trigger more crashes.

The fitness function is the basis for guiding the testcase generation and testcase filtering. General fitness functions aim to produce testcases with better code coverage or closer to potential vulnerability positions. In guiding testcase generation, Li et al. (SP25) and Zhao et al. (SP39) thought that not all codes in the program are vulnerable. Therefore, a neural network model was built to predict which parts were more vulnerable. This guidance was then used to select samples that tended to reach vulnerable locations to mutate. Joffe and Clark (SP40) had the same idea as Li et al., and they constructed the relationship between the library function call information in execution and crashes as a fitness function, which guided AFL to select samples more likely to produce crash for the next mutation. Sperl and Böttinger (SP27) discovered information related to fuzzing from the power side-channel, and used machine learning-based classifiers for branch detection and branch distance classification to select the next input that requires mutation. Zhang et al. (SP33) modeled the linguistic data and proposed a weight query algorithm based on BN to calculate the probability of state transition when the input is was executed. This probability guided the mutation of the voice command on the Intent Classifier in the voice assistant device. Typically, a symbolic execution expert generates a large number of quality inputs improving coverage, but it is a time-consuming process. He et al. (SP34) learned from tens of thousands of sequences of quality transactions generated by running a scalable symbolic execution expert on real-world contracts. After learning, the learned policy was used to guide the generation of testcases for fuzzing unseen programs.

## Generation-based testcase generation

The challenge to be faced in generation-based testcase generation is how to define input format specification, e.g., using grammar [101], block [102], or model [103]. Commonly traditional methods are manually defined, but creating the input data specification in this way is often time-consuming and cannot include all possible kinds of input formats. Machine learning is mainly used to automatically learn the syntax from a collection of corpus that conforms to the input grammar rules. This method can result in a test sample file with a higher pass rate. According to the type of input format of the target program learned in different papers, we summarized the research work of generation-based testcase generation from the three dimensions of file format, protocol, and compilers, respectively.

The first dimension is the file format. Learn&Fuzz (SP2) is the first attempt to learn the grammar rules of PDF format by the seq2seq model. However, the complex file format contains both textual and non-textual data. Moreover only textual data are learned in Learn&Fuzz. To mitigate the challenge of low variety of testcase generated by Learn&Fuzz, IUST DeepFuzz (SP21) distinguishes pure data stored in a file from the meta-data that describes the file format. Testcases without non-textual data are used for training by using LSTM. Finally, the new testcase will be generated by assembling the generated textual data with the mutated non-textual data. Jitsunari and Arahori (SP24) improved the low coverage of Learn&Fuzz from the perspective of generating instruction sequences. Their essential idea was to train a recurrent neural network model from instruction sequences based on mixed character/token-level. New PDF page stream instruction was generated by the trained model for the testcase to induce positive coverage of the instruction-parsing code. Based on the idea of Learn&Fuzz, Sablotny et al. (SP18) confirmed that the stack RNN model could generate HTML tags that cover more program base blocks. Paduraru and Melemciuc (SP20) proposed a testcase generation model that could support any test program or input file. The model clusters the corpus with different file formats. Treating the corpus of the input file as a series of characters, the generation model of each cluster is learned by seq2seq. However, this approach only learns the basic syntax format of each corpus without taking the complex characteristics of different programs or inputs into account.

The second dimension is the protocol. Proprietary network protocols do not expose their protocol formats, and if fuzzing such protocols, the existing method is usually to manually reverse each field in these protocols, which is labor-intensive and time-consuming. To address this problem, Fan and Chang (SP10) proposed using the seq2seq model to learn the protocol format from massive network traffic, and Gao et al. learned the probability distribution of the character in protocols by seq2seq-attention (SP36). Li et al. (SP26) and Hu et al. (SP11) proposed to learn the structure and distribution of data frames by GAN, respectively. Lai et al. (SP43) proposed an anti-sample algorithm that the maximum probability is compared in the algorithm to determine whether to replace the current data value with the minimum probability data value. An RNN model to express the probability distribution of the protocol data values. SeqFuzzer (SP32) proposed a deep learning model that further studies the temporal features of stateful protocols based on learning the structure of protocol frames, and realizes the generation of testcase for stateful protocols.

There is also some research on testcase generation for compilers and engines. Usually, inputs of compilers are highly structured. DeepSmith (SP15) and DeepFuzz (SP29), respectively, use the seq2seq and LSTM models to learn the syntax and semantics of the programming language from the corpus of real-world code, thus to generate a multitude of test code that conforms to the format of the programming language. For JavaScript(JS) engine, Lee et al. (SP42) proposed a novel algorithm of modeling the hierarchical structures of a JS test case and the relationships among such structures into a sequence of fragments. This relationship is used to generate a new AST subtree to replace the original subtree to generate higher quality testcases.

## Testcase filtering

During the fuzz testing, the PUT needs to execute massive samples, and it is time-consuming and inefficient to execute all the samples with uneven quality. The purpose of the testcase filtering is to select the test input that is more likely to trigger new paths or vulnerabilities from a large number of testcases. The input testcase can be analyzed and classified to determine which testcase should be further executed by using machine learning techniques. For using

code coverage as a filter, Gong et al. (SP5) trained a deep learning model based on samples generated by AFL with labels that will change the state of the program. The trained model could predict whether the sample generated by the new round of AFL could change the program state. For vulnerability or unsafe state oriented filters, Zong et al. (SP41) proposed a CNN approach to predict the reachability of inputs (i.e., miss the target or not) before executing the target program, helping directed grey-box fuzzing filtering out the unreachable ones to boost the performance of fuzzing. Chen et al. (SP38) learned a model of the CPS that can predict the effects of actuator configurations on the physical state. This model can later be used to analyze different potential actuator configurations, and help inform which of them is likely to drive the system closer to a targeted unsafe state.

Some studies filter testcases by improving fitness function. Sun et al. (SP16) combined Markov-chain and PCFG model to learn the commonness from a corpus of normal scripts developed by programmers, and used the learned information to compute the script's uncommonness by measuring the deviation of the script to a common script. Scripts with larger deviation may be more likely to trigger errors in the interpreters. Karamcheti et al. (SP14) mapped program inputs to execution trajectories and sorted the entropy of the execution trajectory distribution. It is based on the assumption that the higher the uncertainty, the more likely it is to execute a new code path, so the input with the maximal (most uncertain) entropy is selected as the next input.

## Mutation operator selection

Another problem that needs to be answered in mutation-based testcase generation is how to select a mutation operator. The concept of mutation operators in fuzzing is derived from the genetic algorithm. The mutation operator in fuzzing includes operations such as bit flips, byte flips, arithmetic mutations, dictionary, deleting bytes, cloning bytes, and overwriting bytes. Blind mutation of testcases result in serious waste of testing resource, and better mutation strategy could significantly improve the efficiency of fuzzing. Because choosing mutation operators is similar to the problem that reinforcement learning can address, in the case of mutation operator selection, existing research focuses on using reinforcement learning to select better mutation operators.

When reinforcement learning is used for mutation operator selection, the definition of state, action, and reward need to be determined. In fuzzing of the protocol, Becker et al. (SP1) and LEFT (SP10) defined the message type of the protocol as the state. There is also a set of input substrings as state in fuzzing for the file and compiler (SP17, SP19, SP30, SP35). The action defined in fuzzing methods based on reinforcement learning refers to the mutation operator, which is the same as traditional fuzzing. The reward function indicates which behavior is required in the fuzz testing. Some researchers (SP17, SP19, SP30) used the coverage as the reward function, in which Böttinger et al. and Kuznetsov et al. added processing time as a reward function. What is more, Becker et al. (SP1) designed a comprehensive reward function combining the rewards of function invocation, debugging information, and monitoring the response of the host. LEFT (SP22) defined the rewards function based on emulator crashes, information manufacture, and network connectivity.

In addition to reinforcement learning, Karamcheti et al. (SP13) proposed a Thompson Sampling optimization method based on robbers, which can adaptively adjust the mutator distribution in the process of fuzzing a single program. It is determined which mutation operator should be selected by learning the impact of each mutation operator on code coverage. Raj et al.(SP9) devised another mutaion to test computer vision algorithms, they picks a pattern obtained from the convolutional filter layer of an unrelated CNN and adds it to a random location in the image or video frame.

## Exploitability analysis

Vulnerability exploitability means the possibility that vulnerability is exploited by an attacker. It is an inherent property of vulnerability. In fuzzing, there are massive crashes and error messages, but only a few are vulnerabilities. How to find real vulnerabilities from these crashes is a challenge. General tools include! exploitable [24], CRAX [104], and ExpTrace [105], etc. Whether the vulnerability can be exploited or not is closely related to the program itself. Static information of the program and dynamic information during execution can be collected. The relationship between this information and the exploitability of crashes is discovered by machine learning techniques. The methods of machine learning based exploitability analysis are divided into static-based and dynamic-based methods, depending on how the information is collected.

In a static analysis-based approach, ExploitMeter (SP7) uses the Bayesian machine learning algorithm to make initial judgments on the static features extracted from the software. The initial judgments and the exploitability judgments in the fuzzing process are combined to update the final exploitability results.

In a dynamic analysis-based approach, Exniffer (SP8) used support vector machines (SVM) to learn the features extracted from core dump files (generated during crashes) and information from the most recent processor hardware debugging extensions. Zhang and Thing (SP23) generated compact fingerprints for dynamic execution tracking of each crash input based on n-gram analysis and feature hashing. The fingerprint is then fed to an online classifier to build a distinguishing model. Online classifiers allow models to scale well even for large numbers of crashes by incremental learning, while being easy to update for new crashes.

## Analysis of machine learning based fuzzing model

In the current machine learning based fuzzing research work, there is less work to compare the performance of various algorithms systematically. This section summarizes the knowledge of the machine learning model used in fuzzing. It summarizes the following five aspects:

- Selection of machine learning arithmetics

- Pre-processing methods

- Datasets

- Evaluation metrics

- Hyperparameters setting

## Selection of machine learning arithmetics

The input data of the fuzzer can be hexadecimal text, source code, binary string, network packet, and other forms. The PUT also contains complex syntax, semantics, and logical structure. For complex environments of fuzzing, not all machine learning algorithms can be applied to fuzzing, and choosing different machine learning models for the same problem can lead to significant differences in results [106]. It is a challenge to determine which machine learning models are effective.

Table 4 lists the machine learning arithmetics and their distributions used in the fuzzing. The first column manifests the name of the machine learning models. And the second column indicates the category to which the algorithm belongs, including three categories: traditional machine learning, deep learning, and reinforcement learning. The third column lists the literature that uses the arithmetic. Statistical results demonstrate that each traditional machine

**Table 4. Distribution of machine learning arithmetics for fuzzing.**

| Algorithm | Category | Studies |
|---|---|---|
| LR (*Logistic Regression*) | Traditional Machine learning | SP14 |
| NB (*Naive Bayes*) | | SP7 |
| BN (*Bayesian Networks*) | | SP33 |
| SVM (*Support Vector Machines*) | | SP8, SP38 |
| KNN (*K-Nearest Neighbor*) | | SP27 |
| RF (*Random Forest*) | | SP23 |
| DT (*Decision Tree*) | | SP7 |
| PCFG (*Probabilistic Context-Free Grammar*) | | SP16 |
| PCSG (*Probabilistic Context-Sensitive Grammar*) | | SP6 |
| PA (*Passive-Aggressive*) | | SP23 |
| Thompson Sampling | | SP13 |
| RLS(*Recursive Least Square*) | | SP44 |
| RNN (*Recurrent Neural Network*) | Deep learning | SP11, SP18, SP37, SP40, SP43 |
| CNN (*Convolutional Neural Network*) | | SP9, SP11, SP28, SP41 |
| LSTM (*Long Short-Term Memory*) | | SP2, SP3, SP4, SP10, SP15, SP18, SP19, SP21, SP24, SP31, SP32, SP36, SP38, SP42 |
| GRU (*Gate Recurrent Unit,*) | | SP18, SP34 |
| BLSTM (*Bidirectional Long Short-Term memory*) | | SP4, SP21, SP39 |
| Seq2seq (*Sequence-to-sequence*) | | SP2, SP4, SP10, SP20, SP29, SP32, SP37 |
| seq2seq-attention (*Sequence-to-sequence-attention*) | | SP36 |
| MLP (*Multilayer Perceptron*) | | SP5, SP12 |
| GCN (*Graph Convolutional Network*) | | SP34 |
| Struc2vec | | SP25 |
| GAN (*Generative Adversarial Networks*) | | SP3, SP11 |
| WGAN (*Wasserstein Generative Adversarial Networks*) | | SP12, SP26 |
| Q-Learning | Reinforcement learning | SP22 |
| SARSA (*State–action–reward–state–action*) | | SP1 |
| Deep Q-Learning | | SP17, SP30, SP35 |
| Deep Double Q-Learning | | SP19 |

learning arithmetic is used only once. The reason for this phenomenon may be that traditional machine learning techniques require manual extraction of features. However, both the input sample format contains complex syntactic and semantic structures, and there are no valid vulnerability models or vulnerability features.

Deep learning relies on its representation learning to have the capability to automatically extract features for a wide range of applications in fuzz testing. Two of the most used algorithms are LSTM and seq2seq, which are used 14 times and 7 times, respectively. LSTM is the most commonly used because it is excelling at processing sequential data: the program execution path is very similar to the statement in natural language, and whether a piece of code contains a vulnerability depends on the context. Moreover, LSTM has a memory function for handling long dependencies, which is easy to deal with the code associated with the vulnerability located at a relatively long distance in the path (SP31).

The length of input and output sequences of the seq2seq model is variable, which can use the input of fuzzing as text data effectively to learn local or global syntax information. Besides, other neural networks that have been successfully applied in image processing, such as Generative Adversarial Network [107] and Graph Convolutional Network [108], are also used in the fuzz testing.

Reinforcement learning is used for the selection of mutation operators in fuzzing because reinforcement learning needs to choose different actions in different environments, which is similar to the selection of mutation operators. However, reinforcement learning has its limitations, such as long training time, weak convergence, and local optimization, which leads to its rarely use in fuzzing [109].

## Pre-processing methods

Due to the different types of PUT, the input format is quite different, such as text, pictures, video, network data packets, and program code. However, the input for machine learning is typically vector data, so original input data cannot be fed into the machine learning algorithms directly. Therefore, many methods have been proposed to preprocess the original input data in machine learning based methods. These pre-processing methods can be split into three categories: program analysis, natural language processing, and others. Table 5 summarizes the data preprocessing methods commonly used for fuzz testing.

Program analysis methods extract program features or runtime information, such as stacks, registers, assembly instructions, jumps, program control flow graphs, sequence of function calls and Attributed Control Flow Graph (ACFG), by techniques using static or dynamic analysis (SP8, SP25, SP27, SP34, SP37, SP39, SP40, SP42, SP44). Natural language processing refers to the methods directly letting input as text, using sophisticated text processing techniques to extract hidden features in the input data, such as n-gram [110] (SP7), count statistics (SP6, SP10), one-hot [111] (SP18, SP36), Word2vec [112] (SP31), histogram [113] (SP14), heat map [114] (SP4) and other NLP methods (SP9, SP16, SP33).

In addition, there are some new methods to quantify the input data, for example, combining program analysis with natural language processing techniques (SP23, SP31), encoding the entire input in byte-level or bit-byte (SP4, SP12, SP19, SP24, SP38), mapping characters to decimal (SP11, SP26, SP32, SP35, SP41, SP43), as well as custom methods (SP5). As defined in the literature (SP5), the binary sequence of testcase is represented by 32-bits, fuzzing technique is represented by 4-bit, mutation bits are represented by 10-bit, and the mutation value is represented by 32-bits, and whether the new test case is represented by 1-bit. Finally, each piece of data can be combined into a 79-bit binary sequence, with the first 78-bit as input and the last 1-bit as a label.

**Table 5. Pre-processing methods for machine learning techniques in fuzzing.**

| Pre-processing Method | Description | Studies |
|---|---|---|
| Program analysis | The extracted information is transformed into vectors by static or dynamic analysis. | SP8, SP25, SP27, SP34, SP37, SP39, SP40, SP42, SP44 |
| Natural language processing | Direct use of n-gram, count statistics, Word2vec, heat map, histogram, and other ways to convert the input into a vector. | SP6, SP7, SP9, SP10, SP14, SP16, SP18, SP33, SP36 |
| Others | A combination of program analysis and natural language processing, encode the entire input in byte-level or byte-byte, maps characters to decimal, and custom. | SP1, SP2, SP3, SP4, SP5, SP11, SP12, SP13, SP15, SP17, SP19, SP20, SP21, SP22, SP23, SP24, SP26, SP28, SP29, SP30, SP31, SP32, SP35, SP38, SP41, SP43 |

## Datasets

The performance of machine learning is primarily influenced by the training data. Especially, deep learning can easily lead to over-fitting when the amount of data is insufficient. In the present work, the datasets used for machine learning algorithm-based fuzzing are from the following sources:

- Web-crawler

- Fuzzing generation

- Self-build

- Public dataset

Web crawler [115] is commonly used methods for collecting data, especially for widely used file formats such as DOC, PDF, SWF, and XML. Conventional crawling methods can be downloaded according to specific file extension filter conditions, specific magic bytes, and other signature methods (SP2, SP6, SP34, SP37).

The fuzzing generation is to execute another fuzzer like AFL to collect the generated samples and their labeled data (coverage, code execution path, etc.) for a while. This method can generate datasets in various formats, and the number of samples can be satisfied (SP4, SP5, SP12, SP28, SP36, SP41, SP44).

The self-build approach is similar to the fuzzing generation, but it uses other means. As an example, build a communication environment to grab traffic packets (SP10, SP11, SP26, SP43).

At present, the public dataset used by previous researches and their corresponding categorizations are as follows:

- **Learning from "Big Code" dataset.** [116] The dataset was created by Veselin et al., which is an open-source project consisting of data related to many different programming language codes, such as Python ASTs (This dataset includes 100'000 + 50'000 python files as parsed abstract syntax trees along with the code of the parser) and JavaScript ASTs (This dataset includes 150,000 JavaScript files. The data is available as JavaScript and as parsed abstract syntax trees).

- **The NIST SARD project dataset.** [117] The dataset was published by the National Institute of Standards and Technology (NIST), which has been widely used in many vulnerabilities related work. The dataset contains test cases of over 100,000 different programming languages, covering dozens of different categories of weaknesses, such as those in the Common Weakness Enumeration (CWE).

- **GCC test suite dataset.** [118] The GNU Compiler Collection includes front-ends for C, C++, Objective-C, Fortran, Java, Ada, and Go languages, as well as libraries for these languages (such as libstdc++, libgcj).

- **DARPA Cyber Grand Challenge dataset.** [119] The DARPA Network Challenge Binaries is a set of 200 binary programs with extensive functionalities released by DARPA. These programs are part of an open challenge for creating tools that automatically modify, validate, and fix errors. Common to all of these binaries is that each binary contains one or more bugs, which are generated by humans when they are programming, and it is documented by the developers.

- **LAVA-M dataset.** [120] LAVA-M dataset was published by Dolan-Gavitt et al. The dataset consists of four programs from the GNU Coreutils suite: uniq, base64, md5sum, and who,

which are injected with 28, 44, 57, and 2265 errors with unique IDs, respectively, with some unlabeled errors. These errors are located deep in the program and are only triggered when an offset in the program input buffer matches a 4-byte "magic" (random) value. This dataset has become popular in recent years for benchmarking complex white box fuzzers, symbolic execution tools, and some gray box fuzzers.

- **VDiscovery dataset.** [12] The dataset was released by VDiscovery that is a vulnerability discovery tool and contains a total of 402 unique samples. These samples consist of 138,308 sequences of system calls for 1,039 Debian programs.

## Evaluation metrics

The performance evaluation of the fuzzing methods based on machine learning technology can be divided into two aspects: the evaluation of the performance of the machine learning model and the evaluation of the capability of vulnerability detection.

Since many problems can be treated as classification problems in machine learning based fuzzing, classification metrics are generally used to evaluate the algorithm. Table 5 summarizes the metrics and detailed information used to evaluate the machine learning model in fuzzing. According to the statistics in Table 6, the most commonly used performance metrics are Accuracy, which are closely followed by Precision, Recall, Loss, FPR, and F-measure. FNR and the model perplexity are the least used.

The evaluation of the capability of vulnerability detection for the fuzzing method based on machine learning is the same as that of traditional fuzzing methods. Table 7 summarizes the metrics and details of the vulnerability detection performance that have been used to evaluate the fuzzing method based on machine learning technology.

Both the traditional fuzzing method and the machine learning based fuzzing method are designed to find vulnerabilities. Therefore code coverage, unique crashes, and bugs are valid metrics for evaluating the performance of the fuzzing model. However, the fuzzing method based on machine learning has model training, feature extraction, and other steps, so efficiency is also used in much literature.

**Table 6. Evaluation metrics and details for machine learning models in fuzzing.**

| Performance metric | Description | Studies |
|---|---|---|
| Accuracy | It is the proportion of the total number of correct predictions amongst the total number of correct as well as incorrect predictions. | SP5, SP7, SP21, SP25, SP31, SP39 |
| Precision | It is the proportion of correctly classified fault-prone classes amongst the total number of classified fault prone classes. | SP7, SP8, SP16, SP21, SP31 |
| Recall (true positive rate (TPR)) | It is the proportion of correctly predicted fault prone classes amongst all actual fault-prone classes. | SP8, SP23, SP25, SP31, SP39 |
| FPR (false positive rate) | It is the proportion of all non-fault prone classes which are incorrectly predicted as fault-prone. | SP8, SP23, SP31 |
| FNR (false negative rate) | It is the proportion of faulty classes that are classified as non-fault prone. | SP31 |
| ROC | ROC (receiver operating characteristic curve) is plotted with TPR values on the y-axis and the FPR values on the x-axis. | SP8, SP16 |
| F-measure | It is the harmonic mean of precision and sensitivity. | SP8, SP23, SP31 |
| Loss | It represents some function of the difference between estimated and true values for an instance of data | SP18, SP21, SP25, SP31, SP39 |
| Models perplexity | The perplexity shows the difference between the predicted sequence and test set sequence. | SP21, SP42 |

**Table 7. Evaluation metrics and details for fuzzers based on machine learning.**

| Performance metric | Description | Studies |
|---|---|---|
| Code coverage | It includes instruction coverage, basic block coverage, line coverage, branch coverage, edge coverage, function coverage, and relative coverage. | SP2, SP4, SP6, SP10, SP13, SP14, SP18, SP19, SP20, SP21, SP24, SP25, SP28, SP29, SP30, SP34, SP36, SP37, SP44 |
| Unique code paths | It represents the number of code paths that are executed or triggered during testing. | SP3, SP4, SP12, SP37, SP40 |
| Unique crashes or bugs | It represents the number of unique crashes and bugs found during fuzzing. | SP2, SP4, SP6, SP10, SP15, SP21, SP22, SP25, SP26, SP27, SP28, SP29, SP31, SP32, SP33, SP34, SP37, SP38, SP39, SP40, SP41, SP42, SP43, SP44 |
| Pass rate | It represents the proportion of samples generated that can be verified by the syntax of PUT. | SP2, SP26, SP29, SP32, SP36, SP42, SP43 |
| Efficiency | It represents the measures of time consumption during testing, such as training time of machine learning model, seed generation speed, sample generation speed, execution time, and so on. | SP3, SP5, SP6, SP8, SP10, SP12, SP15, SP17, SP23, SP25, SP26, SP31, SP34, SP35, SP38, SP39, SP41, SP43 |

## Hyperparameters setting

In the implementation of the machine learning model, the value of hyperparameters is not obtained by training but requires artificial settings before training. In general, it is indispensable to optimize the hyperparameters and select an optimal set of hyperparameters to improve the performance and effectiveness of learning. Table 8 summarizes and compares the works of literature and summarizes the values that have been selected for some crucial hyperparameters in machine learning.

The hyperparameters in the deep learning algorithm are mainly selected to complete the comparison, including the number of layers, the number of nodes in each layer, epochs, activation function, and learning rate. The different number of layers and the number of nodes in each layer will affect the accuracy and complexity of the whole neural network. Over-fitting will also occur when the number of nodes in each layer is large. In the fuzzing scenario, the maximum number of layers is 4, and the number of nodes is 128 and 256. As the number of epochs increases, the weight updating iterations of the neural network will increase, and the loss function curve will enter the optimized fitting state from the initial unfitting state to the over-fitting state. Usually, the maximum number of epochs selected is 50, but the best effect can be reached at 40. The choice of the activation function could improve the ability of neural network model expression, and solve the problem that cannot be solved by the linear model. However, the advantages and disadvantages of different activation functions are different, such as sigmoid input range between [0, 1],

**Table 8. Analysis of hyperparameters setting of machine learning models for fuzzing.**

| Hyperparameter | Selected values | Summary |
|---|---|---|
| Number of layers | 1–8 | Select a maximum of 4 layers and 2 hidden layers. |
| Number of nodes in each layer | 32, 64, 100, 128, 256, 512, 1024, 4096, 8192 | Among them, 128 and 256 are more. |
| Epochs | 5, 10, 15, 20, 30, 40, 50 | The maximum epoch's option is 50, but 40 works best. |
| Activation function | Sigmoid, elu, softplus, softsign, ReLU, tanh | Sigmoid, ReLU, and tanh are most commonly used. |
| Learning rate | 0.0001, 0.0005, 0.001, 0.002, 0.02, 0.1 | 0.001 was used the most |

but there are "sigmoid saturate and kill gradients" and not "zero-centered" problems [121]. Tanh [122] address the problem sigmoid output of not "zero-centered", but other problems still existed. In fuzzing, Sigmoid, ReLU, and Tanh are the most commonly used activation functions. The learning rate controls the learning progress of the model and also affects the speed of the model's convergence to the local minimum value. A higher learning rate will easily lead to an explosion and shock of loss value, while a lower learning rate will lead to slow over-fitting and convergence speed. Many values of learning rate are selected in fuzzing, and 0.001 is used more.

## Performance evaluation of fuzzing model based on machine learning

The performance evaluation of machine learning based fuzzing model is divided into two parts. Firstly, we are evaluating the performance of the machine learning model used in fuzzing to discuss whether machine learning technology is with reasonable classification ability in the scenario of fuzz testing. Secondly, we are evaluating the capability of vulnerability detection for the fuzzing model to discuss whether the capability of vulnerability detection is improved by using machine learning technology. However, the detailed parameter settings of the experiments are not given explicitly in many works of literature, and the experimental codes are not open-sourced. Moreover, the types of target programs tested by different fuzzer models are quite different, such as PDF, HTML, various protocols, and compilers. For these reasons, all fuzzer models cannot be evaluated in the same environment and dataset. Therefore, the data used in this section are derived from the experimental data in the corresponding literature. As a whole, these data can reflect the performance of the proposed fuzzer model based on machine learning to some extent.

## Performance evaluation of machine learning arithmetics

In this section, we summarize the results of the machine learning arithmetics. The results in Section 3.4.5 demonstrate that Accuracy, Precision, Recall, and Loss are the most frequently used performance measures in the selected research literature. We provide the results of these four performance measures, which are Accuracy, Precision, and Recall values are selected as the maximum value of the model determined in each literature, the value of Loss is selected as the minimum value of the model determined in the corresponding literature. The results are manifested in Figs 3–6.

The higher value of the three types of performance metrics, including accuracy, precision, and recall, indicates a more accurate prediction. In terms of accuracy, all the models in the listed literature reach a higher value than 0.9. Precision results in Fig 4 also manifest that the machine learning models in the listed literature have precision above 0.92. The statistical results of the performance metrics recall in Fig 5 demonstrate that the lowest recall value in the listed literature is 0.6, and the highest recall is 0.98. Especially, the values of the accuracy and precision are 1 in the Exploitmeter proposed by SP7-DT, which is the highest among the experimental results of multiple types of features. However, all of the experimental results of Exploitmeter's average accuracy is 0.9, the precision of the average of around 0.4, and the average recall is 0.2. The accuracy of V-fuzz proposed by SP25 reaches more than 0.9 after the experiment is stable, but its recall value is low, only 60%. The main reason for the occurrence of the values of these metrics in these two models is that the features selected in the paper are not highly correlated with vulnerabilities, so they cannot be effectively used as the prediction of exploitable samples and vulnerability samples. However, the lower value of the performance metrics Loss indicates the higher robustness of the model. The statistical recall value in Fig 6 is at least 0.05, and the highest is 0.53. The data in this part of the model may be biased, but in

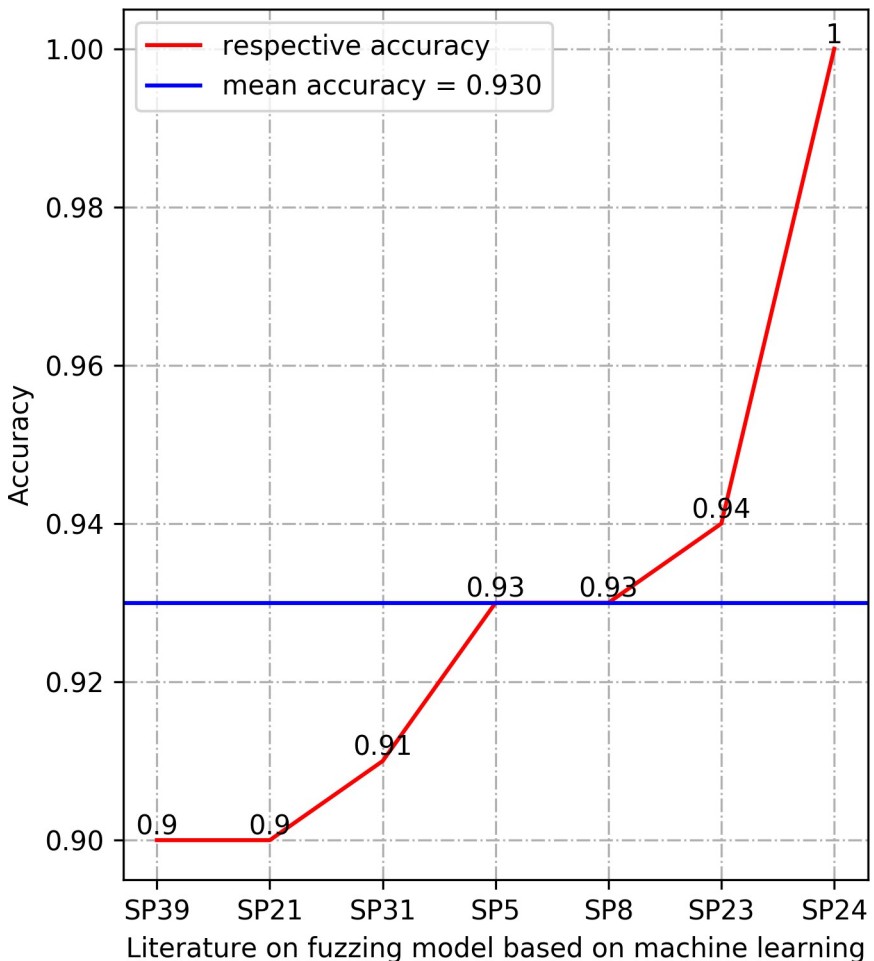

**Fig 3. Comparison of Accuracy between different models.**

the published experimental environment, the values of these performance metrics indicate that the machine learning model for fuzzing has reasonable predictive power.

## Performance evaluation of vulnerability detection

In the fuzz testing, there exist some challenges such as how to mutate and generate input seed files, how to increase coverage rate, and how to pass the validation effectively. Whether the application of machine learning technology can alleviate some bottlenecks effectively is the critical problem studied in this paper. This section summarizes the experimental results of the fuzzing tools based on machine learning and evaluates them from the aspects of code coverage, unique code paths, unique crashes or bugs, pass rate, and efficiency.

**Code coverage.** Coverage is the most common metric used in fuzz testing to evaluate the performance of vulnerability detection. Higher code coverage represents for higher coverage of program execution states, and more thorough testing. Table 9 categorizes and summarizes the experimental results of literature using different code coverage metrics, including basic block coverage, line coverage, relative coverage, function coverage, instruction coverage, and branch coverage. In different dimensions, the results of comparison between machine learning based fuzzer and existing fuzzer are described respectively. Machine learning based fuzzer

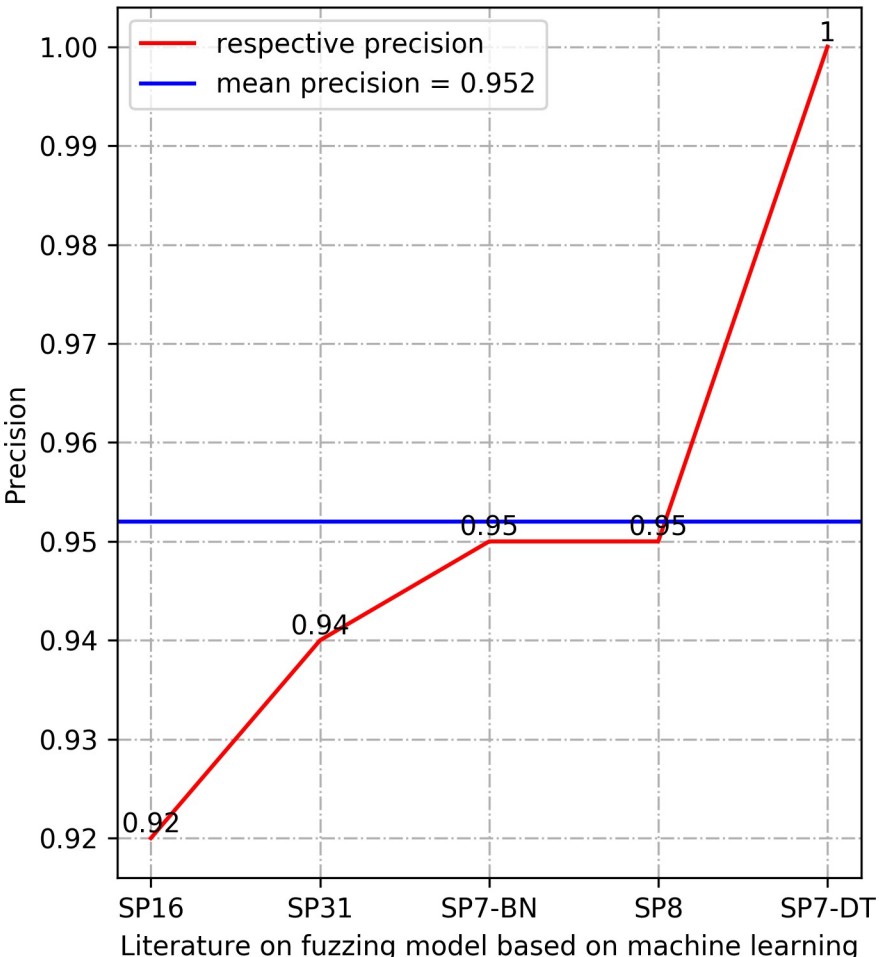

**Fig 4. Comparison of Precision between different models.**

compared to the traditional AFL and its improved version, NEUZZ (SP28) has a fourfold increase in coverage compared to baseline AFL, with a minimum of 1.26% improvement. There is also a minimum 6.69% increase in code coverage compared to other traditional fuzzers, such as libfuzzer [123], Csmith [124], and KleeFL [125]. However, there are also some models (SP25, SP34) that do not improve or even worse in terms of code coverage than traditional fuzzers, but this is only a few phenomena. Moreover, the latest machine learning based fuzzer models, such as Jitsunari'19 (SP24) and Cheng'19 (SP37) improved code coverage compared to the previously proposed machine learning based fuzzer models, such as Learn&Fuzz (SP2) and DeepSmith (SP15). The increase in code coverage of the fuzzer based on machine learning has been improved by a minimum of 0.23% and a maximum of 64 times, with an average increase of 17.3% (when calculating the average value, we exclude the SP28 result because its minimum code coverage increase has reached 2.8 times. If the results are added to the calculation, the average increase in code coverage can reach 3 times, which affects the performance of other models.). On the whole, the application of machine learning technology in fuzzing improves code coverage.

**Unique crashes or bugs.** The purpose of fuzzing is to discover vulnerabilities. The crash or bug that occurs in fuzzing probably contains vulnerabilities, which is the most direct measurement of fuzzer performance. Table 10 summarizes the results of crash and bug

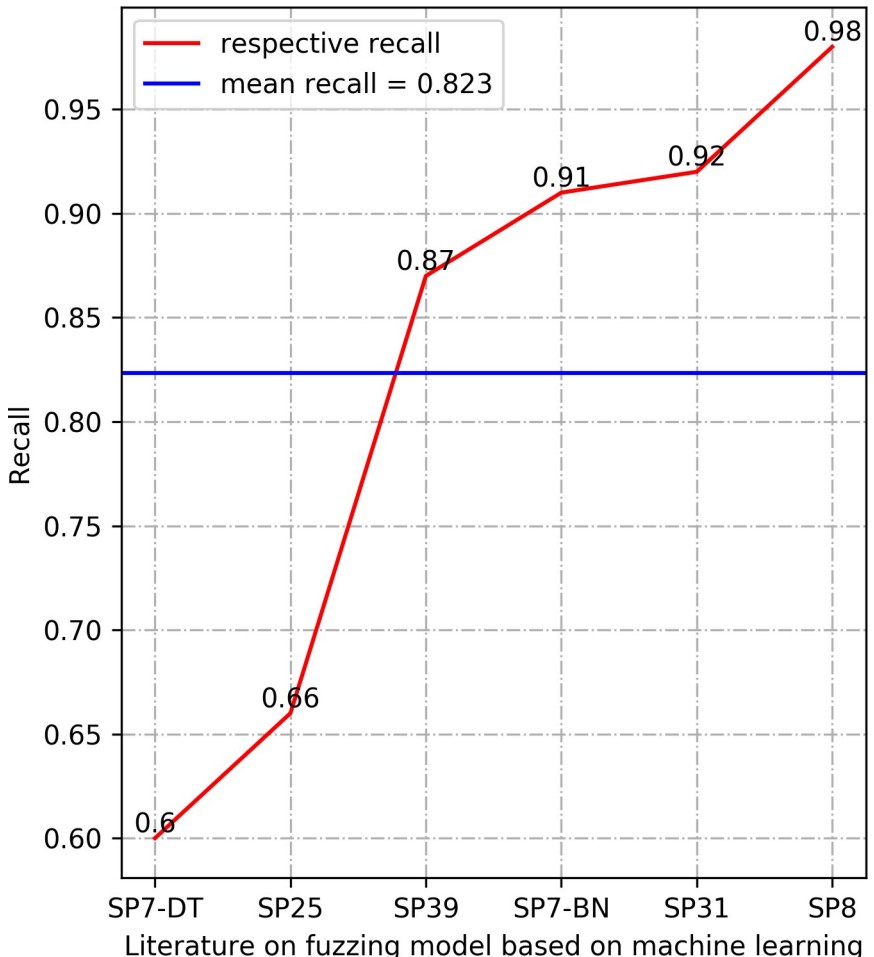

**Fig 5. Comparison of Recall between different models.**

experiments in different works of literature. The first column demonstrates the different fuzzer models based on machine learning. The second column lists the baseline models or methods used in the corresponding literature. And the third column describes the target program tested by fuzzer. The last column summarizes the number of crashes or bugs found in PUT in the fuzzer model based on machine learning compared to different baselines. Machine learning based fuzzers have been found more crashes or bugs than traditional fuzzers, especially the vulnerability-oriented V-fuzz (SP25) is found 3872,14279,14213 more crashes and bugs than baseline Vuzzer, AFL and AFLFast respectively. While this does not suggest that V-Fuzz performs better than another machine learning-based fuzzers, it does indicate that vulnerabilities can be found more effective than baseline tools under the same conditions. Besides, NeuFuzz (SP31), Skyfire+AFL (SP6), Cheng'19 (SP37), SmartSeed+AFL (SP12) models have all published 12, 11, 2, 16 CVE vulnerabilities respectively, which proved the effectiveness of the proposed model. During the fuzz testing, IUST DeepFuzz (SP21) did not find any crash or bug. The main reason was that IUST DeepFuzz did not combine the feedback in the testcase mutation with the fuzzing process, so it could not effectively explore more in-depth code to find a crash or bug.

Table 11 summarizes the number of bugs found by the traditional fuzzing tools and the machine learning based fuzzers on the LAVA-M dataset. The first line lists the four programs

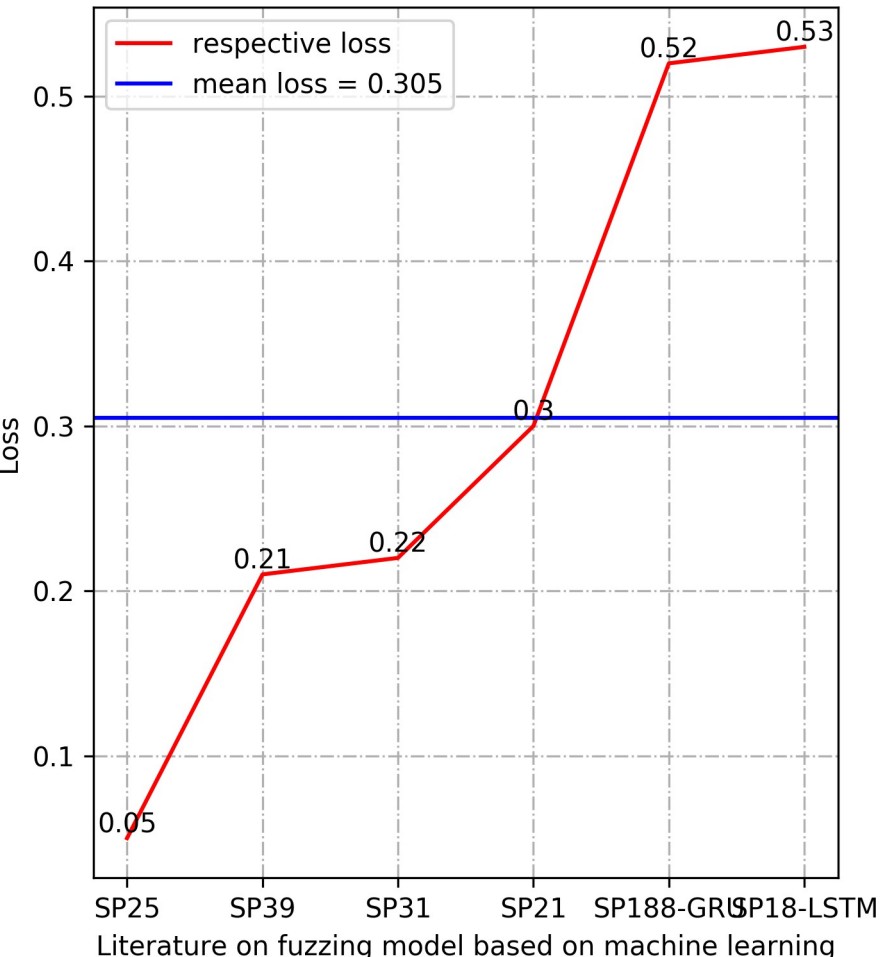

**Fig 6. Comparison of Loss between different models.**

in LAVA-M. The second line lists the number of bugs exposed in each program. The third line to the last line indicates the number of bugs found by each different fuzzers in the four programs of LAVA-M. The last five lines are the experimental results of the fuzzers base on machine learning methods. From the statistical data, it can be found that the capability of vulnerability discovery for the machine learning based fuzzer is not improved compared with the traditional fuzzing tools, such as REDQUEEN [126], DigFuzz [127], Angora [128], InsFuzz [129], and T-fuzz [130]. The only better one is the NEUZZ tool, which can maintain the highest number among three of the four programs. In general, the fuzzing method based on machine learning has not substantially improved the capability of vulnerability detection compared with the traditional fuzzing method. However, there is a threat to the validity of this conclusion. Due to the different input, fewer machine learning based fuzzers can test the LAVA-M dataset, so it is arduous to obtain and summarize the conclusions.

**Pass rate.** Programs often validate the inputs before parsing and handling with magic numbers, magic strings, version number check, and checksums. Invalid testcases are always ignored or discarded in the early stage of execution, and a large number of invalid testcases often result in waste of computing resources and low coverage. The pass rate represents the percentage of generated samples that can pass the validation of the program. Comparing to the 34% pass rate of CFG on XML, Skyfire (SP6) has 85% XSL and 63% XML that can pass

**Table 9. Results of cove coverage improvement of fuzzers based on machine learning.**

| Coverage category | Fuzzing model | Baselines | Results |
|---|---|---|---|
| basic block coverage | SP21 | AFL | + 2.26% [a] |
| | | Augmented-AFL | +7.73% |
| | | Learn&Fuzz | +56% |
| | SP4 | AFL | +1.26% |
| | SP25 | VUzzer | 0 |
| | SP30 | GCC test suite baseline | +37.14% |
| | SP37 | AFL | +2.48% |
| | SP36 | boofuzz | +2.65% |
| line coverage | SP19 | LibFuzzer | +2.5 times |
| | SP6 | Crawl+AFL | +20% |
| | SP29 | Csmith | +6.69% |
| | SP24 | Learn&Fuzz | +0.23% |
| | | DeepSmith | +0.52% |
| relative coverage | SP14 | FidgetyAFL | +11% |
| | | Batched FidgetyAFL | +13% |
| | | Random Batched FidgetyAFL | +6% |
| | SP13 | AFL | +20% |
| | | FidgetyAFL | +11% |
| | | FaireFuzz | +7% |
| function coverage | SP6 | Crawl+AFL | +15% |
| | SP29 | Csmith | +2.26% |
| | SP24 | Learn&Fuzz | +0.46% |
| | | DeepSmith | +0.9% |
| instruction coverage | SP2 | Normal sample execution | +1% |
| | SP10 | Learn&Fuzz | +0.11% |
| | SP34 | Unif | +16% |
| | | Expert | -8% |
| | | Echidna | +24% |
| branch coverage | SP28 | AFL | +4 times |
| | | AFLFast | +2.8 times |
| | | VUzzer | +64 times |
| | | KleeFL | +5 times |
| | | AFL-laf-intel | +54 times |
| | | RNN | +5.7 times |
| | SP29 | Csmith | +2.36% |
| | SP44 | QSYM | +10% |

Remarks: The value after the symbol "+" in the result column indicates the increased coverage of the fuzzing models based on machine learning compared to different baselines. 2) the value after the symbol "-" indicates the decreased coverage of the fuzzing models based on machine learning compared to different baselines.

semantic detection and reach the deeper application execution states due to its consideration of context. Learn&Fuzz (SP2) and Gao'19 (SP36) model produced a sample with a pass rate of more than 96%. The pass rate of DeepFuzz (SP29) increases with the number of iterations. The optimal pass rate of all sampling methods is achieved in 30 iterations of training. The highest pass rate is 82.63%, and the training is stable after 80%. However, the Montage (SP42) sample had a 5.2% lower pass rate than the random selection method, with only 58%. For stateful protocols, SeqFuzzer (SP32) and A-s Fuzzer (SP43) generates more than 90% of the testcase that

**Table 10. Results of Machine learning-based fuzzers compared to baselines in unique crashes and bugs discovery.**

| Fuzzing model | Baselines | PUTs | Crashes or Bugs | |
|---|---|---|---|---|
| SP2 | -- | the PDF viewer included in Microsoft's new Edge browser. | +1 | |
| SP4 | AFL | readpng, readelf, mupdf, and libxml. | +130 | |
| SP31 | PTfuzz | libtiff, binutils, libav, podofo, bento4, libsndfile, audiofile, nasm, and LAVA-M. | +16 | CVE: 12 |
| | QAFL | | +19 | |
| SP6 | Skyfire | Sablotron, libxslt, libxml, and Internet Explorer 11. | +27 | CVE: 11 |
| | Crawl | | +25 | |
| SP15 | CLSmith | OpenCL. | +31 | |
| SP37 | AFL | MuPDF, pdfium, podofo, poppler, and libpng. | +35 (CVE: 2) | |
| SP28 | AFL | readelf, nm, objdump, size, strip, harfbuzz, libjpeg, mupdf, libxml, zlib, LAVA-M, and DARPA CGC. | +21 | |
| | AFLFast | | +33 | |
| | VUzzer | | +53 | |
| | KleeFL | | +46 | |
| | AFL-laf-intel | | +34 | |
| SP25 | Vuzzer | pdftotext, pdffonts, pdftopbm, pdf2svg, MP3Gain, mpg321, xpstopng, xpstops, xpstojpeg, cflow, and LAVA-M. | +3872 | |
| | AFL | | +14279 | |
| | AFLFast | | +14213 | |
| SP12 | Random | MP3gain, ffmpeg, mpg123, mpg321, magick, bmp2tiff, exiv2, sam2p, avconv, flvmeta, ps2ts, and mp42aac. | +755 | CVE: 16 |
| | AFL-result | | +644 | |
| | Peachset | | +608 | |
| | Hotset | | +697 | |
| | AFL-cmin+ AFL | | +676 | |
| SP21 | AFL | MuPDF. | 0 | |
| | Augmented-AFL | | | |
| | Learn&Fuzz | | | |
| SP29 | Csmith | GCC and Clang. | +8 | |
| SP13 | FaireFuzz | DARPA CGC and LAVA-M. | +685 | |
| SP32 | -- | EtherCAT. | +1385 | |
| SP44 | AFL | Tcpdump, binutils-objdump, binutils-readelf, libxml, libtif-tiff2pdff, libtif-tiff2ps, jasper, libjpeg. | +15 | |
| | AFLFast | | +19 | |
| | Angora | | +15 | |
| | QSYM | | +10 | |
| | SavioR | | +7 | |
| SP43 | Kitty Fuzzer | Modbus TCP. | +12 | |
| | Peach Fuzzer | | +16 | |
| SP42 | CodeAlchemist | ChakraCore. | +68 | |
| | Jsfunfuzz | | +76 | |
| | Ifuzzer | | +111 | |
| | Random | | +61 | |
| SP34 | Unif | Ethereum Smart Contracts. | +61 | |
| | Maian | | +86 | |
| | ContractFuzzer | | +184 | |
| SP41 | AFLGo | Bento4, Ettercap, GraphicsMagick, ImageMagick, Jasper, Libming, Libtiff, Libxml2, Podofo, Tcpreplay. | +23 (CVE: 4) | |
| SP38 | Random | the Secure Water Treatment (SWaT) testbed, Water Distribution (WADI) testbed. | +2 | |

Remarks: 1) The value after the symbol "+" in the crashes or bugs column represents the number of crashes or bugs found in the fuzzing models based on machine learning compared to different baselines. 2) The value after the CVE symbol in parentheses represents the number of CVE vulnerabilities discovered by the fuzzing models based on machine learning.

**Table 11. Statistics on the number of vulnerabilities found on LAVA-M dataset by machine learning-based fuzzers and traditional fuzzers.**

|  | base64 | md5sum | uniq | who |
|---|---|---|---|---|
| #Bugs | 44 | 57 | 28 | 2136 |
| AFL | 0 | 0 | 9 | 1 |
| VUzzer | 17 | -- | 27 | 50 |
| FUZZER | 7 | 2 | 7 | 0 |
| SES | 9 | 0 | 0 | 18 |
| Steelix | 43 | 28 | 24 | 194 |
| Angora | 48 | 57 | 29 | 1,541 |
| AFL-laf-intel | 42 | 49 | 24 | 17 |
| InsFuzz | 48 | 38 | 11 | 802 |
| T-fuzz | 43 | 49 | 26 | 63 |
| REDQUEEN | 44 | 57 | 28 | 2134 |
| DigFuzz | 48 | 59 | 28 | 167 |
| SP28 | 48 | 60 | 29 | 1,582 |
| SP31 | 6 | -- | 5 | 8 |
| SP13 | 31 | 1 | 0 | 106 |
| SP25 | 27 | -- | 28 | 62 |
| SP39 | 18 | -- | 5 | 40 |

satisfies the state transition relationship of industrial protocol. As a whole, sample files generated by deep learning and automatic learning based on grammatical semantic information have a high sample pass rate. Nevertheless, finding a security vulnerability requires executing a file whose format is corrupted, so it is necessary to weigh the proportion of sample validity.

**Efficiency.** The test efficiency is to evaluate the time overhead of the fuzzing runtime, including two aspects: execution time and generation time. Execution time indicates the time overhead of executing test samples in the fuzzing process. For example, the time that it needs to find a given number of crashes and hangs, the number that test samples executed per second, and the time that it takes to execute the same test samples. The generation time refers to the time it takes to generate a seed file or a test sample. Table 12 summarizes the experimental results of the test efficiency of the machine learning-based fuzzing models.

Performance metrics in execution time, machine learning based fuzzers are significantly more efficient in execution than traditional fuzzers or common auxiliary methods, with at least 5% (SP5). Especially, DeepSmith (SP15) is 4.46 times better than CLSmith. The main reason is that the fuzzer based on machine learning reduced the execution of a large number of invalid samples. However, there are also cases where execution time is more expensive, A-s Fuzzer is 4% slower than Kitty Fuzzer, and Neufuzz (SP31) is 8% slower than PTfuzz [131] due to the need for path recovery, but these machine learn-based fuzzers take less time than the improved baseline fuzzer. Overall, while the prediction process of the machine learning model introduced into the fuzzing takes a portion of the time, the implementation efficiency can be greatly improved due to the preferential selection of higher quality samples and reduced execution of useless samples.

In terms of generation time, seed or testcase generation based on random selection is the fastest. However, machine learning based seed or testcase generation is more efficient than other strategies, and Faster Fuzzing produces seed files 14.23% quicker than the random approach. In addition, the other models improved by 2.45 times compared to the baseline (SP15) and even 2,000 times (SP12). The results of Skyfire are not summarized in Table 11 because it is not compared to other models, but its execution time and generation time are

**Table 12. Results of efficiency improvement of fuzzers based on machine learning.**

| Efficiency | Fuzzing model | Baselines | Results |
|---|---|---|---|
| Execution time | SP5 | AFL | +5% |
| | SP31 | QAFL | +2.5 times |
| | | PTFuzz | -8% |
| | SP17 | Random selection | + 11.3% |
| | SP15 | CLSmith | +4.46 times |
| | SP43 | Kitty Fuzzer | -4% |
| | | Peach Fuzzer | +46% |
| | SP34 | Unif | +75% |
| | | Echidna | +17% |
| | | Expert | +99 times |
| | SP39 | VUzzer | +41% |
| | SP41 | AFLGo | +5.4 times |
| | SP38 | Random | +21% |
| | SP35 | Random | +30% |
| Generated time | SP3 | Random reinitialization | +14.23% |
| | | LSTM | +60.72% |
| | SP9 | Random | +6 times |
| | SP12 | Peachset | +41 times |
| | | AFL-cmin | +14 times |
| | | Hostset | +2000 times |
| | | Random selection | -480 times |
| | | AFL-result | -480 times |
| | SP15 | CLSmith | +2.45 times |

Remarks: 1) the value after the symbol "+" indicates the efficiency improvement of the fuzzing models based on machine learning compared with different baselines.2) the value after the symbol "-" indicates the decrease of the efficiency of the fuzzing models based on machine learning compared with different baselines.

within the efficient range of 920, 938, and 1008 XSL, XML, and JavaScript per second, respectively. It can execute 3.5, 3.6, and 97.3 XSL, XML, and JavaScript per second, respectively. Compared with other strategies other than random selection, the grammar of the program input is learned by the machine learning method, instead of requiring time-consuming processes such as checking and filtering.

## Discussion

The publication trend indicates that there is an increasing interest in applying machine learning to fuzz testing. However, given the importance of the topic and the relatively small number of studies that have been found, more research is needed in this area. Overall, this systematic review of the 43 studies was helpful in answering our nine research questions.

### RQ1: Why machine learning techniques can be used for fuzzing?

The use of machine learning techniques in fuzzing requires certain prior conditions: 1) training requires massive samples, 2) supervised learning requires the labeled data, and 3) the inputs require to be converted to vectors. For the first and second conditions, the fuzzing process is sufficient because the fuzzing can produce a large number of test samples and crash samples, which can be labeled during sample generation (e.g., whether code coverage increases during execution). Since many of the input files of fuzzing can be treated directly as textual

data, natural language processing provide an effective means of converting various data into vectors. Therefore, the third prior condition is also satisfied. The satisfaction of these prior conditions and the advantages of machine learning have led to rapid growth in the research of machine learning based fuzzing.

### RQ2: Which steps in the fuzzing have used machine learning techniques?

Machine learning techniques have been used in the five stages of fuzzing: seed file generation, testcase generation, testcase filtering, mutation operator selection, and exploitability analysis. In the seed file generation, machine learning technology is mainly used to learn the grammatical and semantic information of the input seed file, the mapping relationship between the input samples and the execution path, or the experience of the existing seed scheduling strategy. In testcase generation, there are two types of testcase generation: mutation-based and generation-based. In the mutation-based method, the use of machine learning technology is mainly to solve where the seed file should be mutated; In the generation-based method, machine learning technology learns mainly the grammatical and semantic information of input samples to generate samples that conforms to the input grammar rules. In testcase filtering, machine learning is used to build a classifier, so that it can distinguish which testcases are easier to execute more code and trigger crashes more easily. In mutation operator selection, the most commonly used machine learning algorithm is reinforcement learning, which uses a defined reward mechanism to guide the selection of mutation operators during fuzzing. In exploitability analysis, machine learning is used to learn the relationship between the static or dynamic features of the program and the crash availability, so as to determine whether the crash is available.

Among these steps, machine learning is the most used in the testcase generation step. One of the main reasons is that training samples are easier to collect in this step. Since there is no baseline for comparison, we are not sure in which step using machine learning will achieve better performance, but these steps of using machine learning for fuzzing have improved fuzzing performance.

### RQ3: Which machine learning algorithms have been used for fuzzing?

Traditional Machine learning, deep learning, and reinforcement learning have been used in fuzzing. However, deep learning algorithm is more widely used due to its robust learning and expression ability. Among the machine learning algorithms, the two most used algorithms are LSTM and seq2seq. At present, the emerging graph convolutional neural network is also used in fuzzing.

### RQ4: Which techniques are used for data pre-processing of fuzzing based on machine learning?

There are three kinds of preprocessing methods for fuzzing based on machine learning: program analysis, natural language processing, and others. Program analysis methods extract program features or runtime information, such as stacks, registers, assembly instructions, jumps, program control flow graphs, sequence of function calls and Attributed Control Flow Graph (ACFG), by techniques using static or dynamic analysis; Natural language processing refers to the methods directly letting input as text, using sophisticated text processing techniques to extract hidden features in the input data, such as n-gram, count statistics, one-hot, Word2vec, histogram, heat map, and other NLP methods; Others refer to the combination of program analysis and natural language processing technology, or encode the entire input into byte-level or bit-byte, or map characters to decimal, etc.

### RQ5: Which datasets are used for training and evaluating?

The datasets used for machine learning mainly come from four categories: web-crawlers, fuzzing generation, self-build, and public datasets. Web crawlers refer to collecting data in corresponding formats from the Internet, which is a relatively common method. The fuzzing generation is to execute another fuzzer like AFL to collect the generated samples and their labeled data (coverage, code execution path, etc.), which is currently the most used method for constructing datasets. The self-build approach is similar to the fuzzing generation, but it uses other means, which is usually used to build protocol datasets. There are fewer public datasets, among which the most used ones are the Learning from the "Big Code" dataset, the NIST SARD project dataset, the GCC test suite dataset, the DARPA Cyber Grand Challenge dataset, LAVA-M dataset, and VDiscovery dataset.

### RQ6: Which performance measures are used for evaluating the results?

The most commonly used metrics for evaluating machine learning algorithms in fuzzing are accuracy, precision, and recall, which are closely followed by Recall, Loss, FPR, and F-measure.

### RQ7: How to set the hyperparameters of the machine learning arithmetic?

The hyperparameters in the deep learning algorithm are mainly selected to complete the comparison, including the number of layers, number of nodes in each layer, epochs, activation function, and learning rate. In the fuzzing scenario, the maximum number of layers is 4, the number of nodes is 128 and 256, and the maximum number of epochs selected is 50, but the best effect of most models can be achieved in 40 epochs. Sigmoid, ReLU, and Tanh are the most commonly used activation functions. Many values of learning rate are selected in fuzzing, and 0.001 is used more.

### RQ8: What is the performance of the machine learning arithmetics?

The performances of the machine learning arithmetics are assessed based on four metrics: accuracy, precision, recall, and loss. Only 5 to 7 out of the 44 primary studies have evaluated the performances of machine learning algorithms. According to the statistical analysis of the existing literature, the average value of the machine learning arithmetics for fuzzing in the four measurement dimensions of accuracy, precision, recall, and loss reaches 0.93, 0,952, 0.823, 0.305, respectively. Although the results may not be comprehensive, we can find from the results of the existing data that the current machine learning algorithms used for fuzzing have higher classification/prediction capabilities and robustness.

### RQ9: What is the capacity of the machine learning techniques based fuzzers to discover vulnerabilities?

For the performance evaluation of vulnerability discovery of machine learning based fuzzers, we mainly evaluated it from the five dimensions of code coverage, unique code paths, unique crashes or bugs, pass rate, and efficiency. From the evaluation results, we can find the following results:

1. In the dimension of code coverage, the fuzzers based on machine learning was improved by an average of 17.3% on the code coverage compared to the baseline (excluding SP28 results).

2. In the dimension of unique code paths, the machine learning based fuzzers have a minimum increase of 6.16% and a maximum of 2 times, an average increase of 56.83% compared to the baseline.

3. In the dimension of unique crashes or bugs, the machine learning-based fuzzers can find more unique crashes and bugs on the real-world more than the baseline, but the results of the bugs found on the LAVA-M dataset are poor. On the one hand, the reason is that there may be less statistical data, and the analysis is not comprehensive. On the other hand, the vulnerabilities on LAVA-M dataset are hidden deeper. Traditional fuzzers have better program analysis capabilities than fuzzers based on machine learning, so they can bypass more complicated magic byte check to trigger these bugs.

4. In the dimension of pass rate, the minimum pass rate of testcases generated by machine learning based fuzzers can reach 58%, the highest can reach 99.99%, and the average pass rate is 83.95%.

5. In the dimension of efficiency, the fuzzers based on machine learning have a minimum execution time improvement of 5% and a maximum of 99 times compared to the baseline, and a minimum increase of 14.23% and a maximum of 2000 times in testcase generation time.

In terms of the evaluation results of each dimension, there are still some machine learning-based fuzzers that have not improved in performance compared to traditional fuzzers, but the results are slightly worse. However, we can find from the overall analysis of all dimensions: 1) SP25 does not improve coverage compared to Vuzzer, but SP25 finds 3872 more unique crashes than Vuzzer, 2) SP34 has 8% less code coverage than Expert, but its performance in execution time is 99 times better than Expert, 3) SP21 has not found more unique crashes or bugs than AFL, Augmented-AFL, and Learn & Fuzz, but its code coverage has increased by 2.26%, 7.73%, 56%, respectively, 4) The pass rate of the testcase generated by SP42 is 5.2% lower than that of the randomly generated testcase, but the SP42 finds 61 more unique crashes compared to the randomly generated method. 5) SP31 is 8% slower than PTFuzz in execution time, but it finds 16 more crashes compared with PTFuzz, 6) SP12 is 480 times slower than random and AFL-result in generating testcase time, but it finds 755 and 644 more crashes than random and AFL-result, respectively. Through a comparative analysis of the vulnerability discovery performance of machine-based fuzzers in these five dimensions, it can be concluded that machine-based fuzzers improve the performance of vulnerability discovery compared to traditional fuzzers.

## Limitations

This systematic review considered a number of primary studies to evaluate and assess the performance of various machine learning techniques amongst themselves. A limitation in this review is that the literature is insufficient. Thus, the comparison is not definitely conclusive. Moreover, while comparing the performance measures of fuzzer based on machine learning, each primary study may use different experimental settings, which include datasets, feature reduction methods, weighing methods, and pre-processing methods [132]. Though we have exhaustively searched all the stated digital search libraries, there still may be a possibility that a suitable study may be left out. Also, this review does not include any unpublished research studies [133]. We have assumed that all the studies are impartial, however, if this is not the case then it poses a threat to this study.

## Conclusion and future directions

In this paper, we systematically reviewed the works of literature to analyze and assess the performance of machine learning techniques for fuzzing. First, we introduced the concept of

fuzzing and the key challenges that currently exist. Second, we analyzed and summarized the reasons why machine learning technology can be used in fuzzing scenarios. Third, we emphatically summarized the use of machine learning techniques for different stages. Fourth, we summarized the characteristics of the primary research from the perspectives of selection on machine learning algorithms, pre-processing methods, datasets, evaluation metrics, and hyperparameters settings. Then the performances of the machine learning for fuzzing are assessed based on four metrics (accuracy, precision, recall, and loss). The results depict that machine learning has an excellent predictive capability for fuzzing. Finally, the capability of vulnerability discovery for the machine learning based fuzzer is analyzed by comparing it with the traditional fuzzer. By comparing code coverage, unique code paths, unique crashes or bugs, pass rate, and efficiency, the results depict that the introduction of machine learning technology in fuzzing can improve the performance of fuzzing.

Future directions can be carried out from the following four aspects:

## Datasets

In this study, we find that there is no generally accepted dataset that can be used as a benchmark in the current fuzzing field. Dataset constructed from web crawlers, fuzzing generation, self-builds are not universal and are less recognized. Some public datasets have been used to contain fewer categories, fewer features, and the data used for training is unbalanced. The quality of the dataset will seriously affect the performance of the vulnerability detection model. Therefore, we believe that the introduction of machine learning into fuzzing must establish an open dataset that can be used as a test benchmark.

## Feature selection

Machine learning constructs a classification model by learning the characteristics of the dataset. The selection of different features will lead to different classification accuracy and precision of the model. The structure of the program and the execution information are not directly related to the vulnerability in the field of vulnerability discovery, so how to select practical features from the program or sample becomes an essential factor affecting the performance of fuzzing. At present, the natural language processing technology is relatively mature, so we can consider using advanced technologies in the field of natural language processing to extract useful information, such as code attributes, semantic and grammatical features of programs for fuzzing.

## Selection of learning algorithms

Different machine learning techniques are suitable for different scenarios, and different network configurations can lead to different results. First of all, the characteristics of different stages of fuzzing, the size of the corresponding data, and the advantages and disadvantages of different algorithms should be used as the basis for the algorithm selection. Secondly, graph convolutional networks, fusion neural networks, and interpretable deep learning models can all be tried to integrate with fuzzing, and it is necessary to study more complicated and suitable neural network models to improve the quality of generated samples.

## Directed grey-box fuzzing

Directed grey-box fuzzing becomes popular in the field of vulnerability discovery. Different from coverage-based fuzzing, they believe that not all codes in the program are vulnerable. The goal of directed grey-box fuzzing is to check whether a piece of potentially buggy code

really contains a bug. How to identify a piece of code as potentially buggy code and convert it into a feature vector as an input to machine learning will be a challenge.

## Supporting information

**S1 File. PRISMA checklist.**
(DOC)

## Author Contributions

**Conceptualization:** Yan Wang, Peng Jia, Luping Liu, Cheng Huang, Zhonglin Liu.

**Data curation:** Yan Wang.

**Formal analysis:** Yan Wang, Peng Jia, Luping Liu.

**Investigation:** Yan Wang, Zhonglin Liu.

**Writing – original draft:** Yan Wang, Zhonglin Liu.

**Writing – review & editing:** Peng Jia, Luping Liu, Cheng Huang.

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
