## [Decision Letter · Decision Letter 0]

8 Jul 2020

PONE-D-20-15933

A systematic review of fuzzing based on machine learning techniques

PLOS ONE

Dear Dr. Liu,

Thank you for submitting your manuscript to PLOS ONE. After careful consideration, we feel that it has merit but does not fully meet PLOS ONE’s publication criteria as it currently stands. Therefore, we invite you to submit a revised version of the manuscript that addresses the points raised during the review process.

We look forward to receiving your revised manuscript.

Kind regards,

Tao Song

Academic Editor

PLOS ONE

Journal Requirements:

2) PLOS requires an ORCID iD for the corresponding author in Editorial Manager on papers submitted after December 6th, 2016. Please ensure that you have an ORCID iD and that it is validated in Editorial Manager. To do this, go to ‘Update my Information’ (in the upper left-hand corner of the main menu), and click on the Fetch/Validate link next to the ORCID field. This will take you to the ORCID site and allow you to create a new iD or authenticate a pre-existing iD in Editorial Manager. Please see the following video for instructions on linking an ORCID iD to your Editorial Manager account: https://www.youtube.com/watch?v=_xcclfuvtxQ

3)  Thank you for stating the following in the Acknowledgments Section of your manuscript:

[This work was supported by the National Key Research and Development Program of China under Grant

2017YFB0802900.]

 [The author(s) received no specific funding for this work.]

Please include the updated Funding Statement in your cover letter. We will change the online submission form on your behalf.

4) In your Data Availability statement, you have not specified where the minimal data set underlying the results described in your manuscript can be found. PLOS defines a study's minimal data set as the underlying data used to reach the conclusions drawn in the manuscript and any additional data required to replicate the reported study findings in their entirety. All PLOS journals require that the minimal data set be made fully available. For more information about our data policy, please see http://journals.plos.org/plosone/s/data-availability.

5) Please include captions for your Supporting Information files at the end of your manuscript, and update any in-text citations to match accordingly. Please see our Supporting Information guidelines for more information: http://journals.plos.org/plosone/s/supporting-information.

Reviewers' comments:

Reviewer's Responses to Questions

**Comments to the Author**

1. Is the manuscript technically sound, and do the data support the conclusions?

Reviewer #1: Yes

2. Has the statistical analysis been performed appropriately and rigorously? 

Reviewer #1: Yes

3. Have the authors made all data underlying the findings in their manuscript fully available?

Reviewer #1: Yes

4. Is the manuscript presented in an intelligible fashion and written in standard English?

Reviewer #1: Yes

5. Review Comments to the Author

Reviewer #1: The authors review the research progress of using machine learning techniques for fuzz testing in recent years, analyzes how machine learning improves the fuzzing process and results, and sheds light on future work in fuzzing. The results depict that the introduction of machine learning techniques can improve the performance of fuzzing. This work could provide researchers with a systematic and more in-depth understanding of fuzzing based on machine learning techniques and provide some references for this field through analysis and summarization of multiple dimensions.

6. PLOS authors have the option to publish the peer review history of their article (what does this mean?). If published, this will include your full peer review and any attached files.

Reviewer #1: No

---

## [Author Response · Author response to Decision Letter 0]

11 Jul 2020

Dear Professors,

 We owe a lot of thanks to the editors and reviewers for your reading our paper with great care and for your helpful comments. The comments are all very instructive. We have performed revisions to the manuscript according to the constructive suggestions, and we believe that the paper has been improved much. Detailed explanations are attached below. We hope it would be acceptable for publication.

Thank you again for your time and consideration.

Yours sincerely

Yan Wang, Peng Jia, Luping Liu, Cheng Huang, and Zhonglin Liu

Answers to the Comments and Revision Explanations

Academic Editor:

Comment 1: Please ensure that your manuscript meets PLOS ONE's style requirements, including those for file naming.

Answer and Explanation 1: 

Thanks for your advice! We reviewed the format descriptions in the PLOSOne_formatting_sample_main_body.pdf and PLOSOne_formatting_sample_title_authors_affiliations.pdf documents, and referred to articles published in PLOS ONE. We have modified the format of author byline, font size, the format of headings, the format of the table, the figure citations, Finally, the name of the supporting file was modified. We used the "Track Changes" option in Microsoft Word to show these changes.

Comment 2: PLOS requires an ORCID iD for the corresponding author in Editorial Manager on papers submitted after December 6th, 2016. Please ensure that you have an ORCID iD and that it is validated in Editorial Manager. To do this, go to ‘Update my Information’ (in the upper left-hand corner of the main menu), and click on the Fetch/Validate link next to the ORCID field. This will take you to the ORCID site and allow you to create a new iD or authenticate a pre-existing iD in Editorial Manager. Please see the following video for instructions on linking an ORCID iD to your Editorial Manager account: https://www.youtube.com/watch?v=_xcclfuvtxQ

Answer and Explanation 2: 

Thanks for reminding us! The corresponding author has created the ORCID iD and authenticated it on the Editorial Manager.

Comment 3: Thank you for stating the following in the Acknowledgments Section of your manuscript:

[This work was supported by the National Key Research and Development Program of China under Grant 2017YFB0802900.]

[The author(s) received no specific funding for this work.]

Please include the updated Funding Statement in your cover letter. We will change the online submission form on your behalf.

Answer and Explanation 3:

Thanks for reminding us! In response to your question, I have deleted the Acknowledgements part in the manuscript.

Comment 4: In your Data Availability statement, you have not specified where the minimal data set underlying the results described in your manuscript can be found. PLOS defines a study's minimal data set as the underlying data used to reach the conclusions drawn in the manuscript and any additional data required to replicate the reported study findings in their entirety. All PLOS journals require that the minimal data set be made fully available. For more information about our data policy, please see http://journals.plos.org/plosone/s/data-availability.

Answer and Explanation 4:

Thanks for reminding us! Our data mainly comes from public papers. We have uploaded the 44 papers finally selected in this manuscript to GitHub, and the URL accessed is: https://github.com/wtwofire/A-systematic-review-of-fuzzing-based-on-machine-learning-techniques

Comment 5: Please include captions for your Supporting Information files at the end of your manuscript, and update any in-text citations to match accordingly. Please see our Supporting Information guidelines for more information: http://journals.plos.org/plosone/s/supporting-information.

Answer and Explanation 5:

Thanks for your advice. we have added a caption for our Supporting Information files at the end of our manuscript, and update in-text citations to match accordingly.

---

## [Editor Report · Decision Letter 1]

3 Aug 2020

A systematic review of fuzzing based on machine learning techniques

PONE-D-20-15933R1

Dear Dr. Liu,

We’re pleased to inform you that your manuscript has been judged scientifically suitable for publication and will be formally accepted for publication once it meets all outstanding technical requirements.

Kind regards,

Tao Song

Academic Editor

PLOS ONE
---

## [Editor Report · Acceptance letter]

7 Aug 2020

PONE-D-20-15933R1 

A systematic review of fuzzing based on machine learning techniques 

Dear Dr. Liu:

I'm pleased to inform you that your manuscript has been deemed suitable for publication in PLOS ONE. Congratulations! Your manuscript is now with our production department. 

Kind regards, 

on behalf of

Dr. Tao Song 

Academic Editor

PLOS ONE